# Development and Applications of a Pressurized Water-Filled Impedance Tube

**DOI:** 10.3390/s22103827

**Published:** 2022-05-18

**Authors:** Zong-You Shen, Ching-Jer Huang, Kuan-Wen Liu

**Affiliations:** 1Department of Hydraulic and Ocean Engineering, National Cheng Kung University, Tainan 70101, Taiwan; n88051079@gs.ncku.edu.tw; 2Coastal Ocean Monitoring Center, National Cheng Kung University, Tainan 70101, Taiwan

**Keywords:** pressurized water-filled impedance tube (WFIT), three-parameter calibration method (3PCM), reflection coefficient, water–air interface, porous rubber material

## Abstract

In this study, a pressurized, water-filled impedance tube (WFIT) was developed to measure the reflection coefficients of sound-absorbing materials under various hydrostatic pressures. The developed WFIT was calibrated using a two-microphone, three-parameter calibration method (3PCM). The accuracy and repeatability of the measured reflection coefficients for the water–air interface in the WFIT were determined by comparing these coefficients with corresponding theoretical reflection coefficients. The WFIT was then used to measure the acoustic reflection coefficient of a porous rubber specimen on three dates, and the corresponding measurement results exhibited satisfactory repeatability. The aforementioned impedance tube was also used to measure the reflection coefficient of a porous rubber specimen under a hydrostatic pressure of 4 P_atm_ three times on the same day, and one time each on three days, using the same experimental setup and measurement procedure. The results obtained in the aforementioned tests also exhibited satisfactory repeatability. Finally, the WFIT was used to measure the reflection coefficients of porous rubber specimens with various thicknesses under different hydrostatic pressures. The results of this study indicate that the developed WFIT calibrated with the 3PCM can achieve suitable repeatability in the measurement of the reflection coefficients of sound-absorbing materials under various hydrostatic pressures.

## 1. Introduction

Passive and active sonars have been widely used to detect the locations of underwater sound sources or targets. The working principle of active sonar mainly relies on the emission of sound waves from an acoustic device into seawater. The sound waves emitted by a sonar are reflected from a target object, and the reflected waves are received by a sonar receiver. The position of the target is then determined by analyzing the reflected signals using various signal processing techniques, such as triangulation, beamforming, and time-reversal mirror [1]. To improve the stealth performance of a submarine, the sound-absorbing properties of the hull material at various submergence depths must be determined.

The acoustic properties of a material in air are usually measured using an acoustic impedance tube. ASTM E1050-12 [2] describes a standard method for testing the impedance and absorption of a material in air using a tube and two microphones. Commercial products, such as the B&K Type 4206 Impedance Tube, are available for such testing. By contrast, no standard specifications and equipment are available to measure the acoustic properties of materials in water, especially under various hydrostatic pressures. Therefore, the main objective of this study was to develop a pressurized water-filled impedance tube (WFIT) that can be used to measure the acoustic properties of materials at different water depths.

Two methods have been developed to measure the acoustic reflection coefficient with an air impedance tube: the standing wave ratio method [3] and transfer function method [2]. The two-microphone transfer function method eliminates the disadvantages of the single-microphone standing wave ratio method, which requires successive movements of the microphone; thus, this method has higher efficiency and accuracy than the single-microphone standing wave ratio method. The transfer function method was first proposed by Seybert and Ross [4], and its underlying working principle is described in the following text. Periodic sound waves are generated at one end of an impedance tube, and the test sample is placed at the other end of the tube. Sound pressures are measured using two microphones at various tube locations. The ratio of the sound pressures at two tube locations is defined as the transfer function. The acoustic reflection coefficient is then obtained after the separation of the incident and reflected sound waves in the impedance tube.

Chung and Blaser [5,6] proposed a method similar to the transfer function to measure the acoustic reflection coefficient. They decomposed broadband random waves into incident and reflected components using the transfer function relation between the sound pressures at two tube positions. A sensor-switching technique based on the transfer function method has been proposed to eliminate signal distortion and improve system calibration.

The sensor-switching technique was first applied by Corbett [7] to study the acoustic properties in a WFIT. The amplitude (modulus) and phase angle of the measured complex reflection coefficient at the interface between water and air differed from the corresponding theoretical values by 7% and 30°, respectively. Wilson et al. [8] proposed a two-microphone, three-parameter calibration method (3PCM) for a WFIT to reduce measurement errors, such as acoustic errors, microphone sensitivity errors, and signal-receiving errors. They examined the elastic waveguide effect inside the tube and the influence of hydrophones on acoustic transmission in water. In addition, Jones and Stiede [9] compared different modes of acoustic sources in an impedance tube, such as mono-frequency sound and white noise, and concluded that higher accuracy is achieved for reflection measurement with mono-frequency acoustic sources than with white noise. Bodén and Åbom [10] conducted a systematic investigation of various errors that might occur in the measurement of reflection coefficient for a material in an air-filled impedance tube using the two-microphone transfer function method.

The sound waves in a WFIT are assumed to be plane waves, and materials with relatively poor sound absorption properties are selected as the tube wall materials for a WFIT. However, the medium in a WFIT is water, whose characteristic impedance is close to that of the tube wall (which is usually made of stainless steel). Accordingly, the effects of the elastic waveguide in a WFIT influence the sound wave propagation in the tube. Del Grosso [11] proposed a theoretical model for the effects of elastic waveguides and obtained the radial particle displacements for various wave-propagation modes in a WFIT. He indicated that in the zeroth-order mode, the sound wave transmission in a WFIT approximates the transmission of plane waves. Moreover, Del Grosso found that the particle displacement and wavefront curvature of the first-order mode were 20 and 14 times those of the zeroth-order mode, respectively. Lafleur and Shields [12] validated the theoretical model of Del Grosso [11] by conducting experiments on various tube wall materials. Wilson [13] and Jian [14] have studied the effect of wall thickness on sound transmission by comparing the numerical and experimental results obtained for the reflection coefficient, and have verified the feasibility of plane wave transmission.

In experiments conducted with a WFIT, air is entrained into the water column when filling the tube with water. The speed of sound waves in a liquid is considerably affected by the number and size of bubbles in the liquid [15,16]. Therefore, the existence of bubbles in the water in a WFIT considerably increases the measurement uncertainty. To minimize this effect, Wilson et al. [8] filled a WFIT with distilled and degassed water several hours before conducting experiments.

Wilson et al. [8] verified the accuracy of the measured reflection coefficient at the free surface in a WFIT when using the 3PCM to calibrate this tube; however, they did not measure the reflection coefficient of sound-absorbing materials in a WFIT. Zhou et al. [17] reported that the measured reflection coefficient at the free surface in a WFIT was closer to the corresponding theoretical value when the 3PCM was used than when the transfer function method was used. In addition, measurements with a higher repeatability were obtained for the reflection coefficient of sound-absorbing materials when the 3PCM was used than when the transfer function method was used.

Fu et al. [18] measured the sound absorption coefficient of silicon-based rubber with steel plate backing in a WFIT under various hydrostatic pressures by using the transfer function method, without employing the 3PCM. Repeatability tests revealed that the mean variation in the measured sound absorption coefficients was within 5%. However, the variations reached up to 20% at certain frequencies. The results of the aforementioned tests indicated that measurements of the sound absorption coefficient are highly sensitive to the installation and mounting conditions of the sample.

Sun and Hou [19] used a WFIT and the pulse separation method to measure the reflection coefficient of sound at the water–air interface in the tube and the absorption coefficient of a sound-absorbing material. Their results reveal that the amplitude and phase errors of the reflection coefficients at the water–air interface were 5% and 5°, respectively. The tested sound frequencies in [19] ranged from 500 Hz to 2 kHz. On the basis of the ASTM E2611 standard [20], Oblak et al. [21] developed a four-microphone impedance tube for measuring the sound reflection and transmission of porous material in liquids in a low-frequency range. This device was used to obtain the transmission loss of low-frequency sounds (100–500 Hz) passing through metal-foam samples.

Sun and Hua [22] used the finite element (FE) software ANSYS to simulate the absorption coefficient of a sound-absorbing material installed in a WFIT. The FE results were verified by comparing them with the analytical solutions obtained using the standing wave ratio method, transfer function method, and pipe pulse method. After verification of the FE results, the FE algorithm and transfer function method were employed to investigate the system errors caused by various experimental factors, including the size of the slit between the side face of the sample and the inner wall of the steel tube, the inclination of sample, and the surface finish of the sample.

In addition to the impedance tube, which is widely used to measure the reflection coefficient of a material (usually a solid-state material), the pulse echo technique was developed to determine the acoustic impedance of a liquid by measuring the ultrasonic velocity in the liquid and its liquid density. Joshi et al. [23,24] and Iwase et al. [25] have described this technique in detail. Furthermore, ultrasound-based elastography methods were developed to determine the bulk modulus or density of hard and soft materials [26].

Wilson et al. [8] revealed that the 3PCM considerably improves the repeatability and accuracy of measurements in a WFIT; however, this method has not been widely employed for calibrating WFITs [14,18,21]. Furthermore, a pressurized WFIT is required to determine the behavior of sound-absorbing materials at various water depths; however, few studies have investigated the repeatability of measurements conducted using a pressurized WFIT.

On the basis of the standard test method described in ASTM E1050-12 [2], a vertical pressurized WFIT was developed in this study to measure the acoustic reflection coefficient of a material under various hydrostatic pressures. This WFIT was calibrated using the two-microphone 3PCM. In addition, because higher accuracy is achieved in reflection measurements when using a mono-frequency acoustic source in a WFIT than when using white noise [9], the NEPTUNE-TX335 transducer (Neptune Sonar Limited, East Yorkshire, the United Kingdom) was used to emit mono-frequency (monochromatic) sounds at frequencies ranging from 2 to 7 kHz in the developed WFIT.

The acoustic reflection coefficient at the water–air interface in the developed WFIT was measured using various methods, including the transfer function method, sensor-switching technique, and 3PCM. The accuracy of the measured values was determined by comparing them with corresponding theoretical values. The results obtained using the 3PCM were more accurate than those obtained using the other methods. Furthermore, the results obtained using the 3PCM had high repeatability. After the accuracy and repeatability of the experimental results obtained at the water–air interface were verified, the developed WFIT was used to obtain the reflection coefficient of a porous rubber specimen under a hydrostatic pressure of 4 P_atm_ to test the repeatability of the measurement results. Subsequently, the WFIT was used to measure the reflection coefficients of porous rubber specimens with various thicknesses under different hydrostatic pressures, such as 1, 4, and 8 P_atm_.

## 2. Theoretical Background

### 2.1. Acoustic Impedance

When sound waves are incident on the interface between two media, reflection and transmission occur at the interface because of the different impedances of these media (Figure 1). The reflection coefficient R, which is defined as the ratio of the reflected wave pressure to the incident wave pressure, is a complex parameter that represents the amplitude ratio and phase difference. If the incident sound waves are plane waves and the second transmission medium has a semi-infinite domain (Figure 1), R can be expressed as follows:(1)$$R=\frac{\rho_{2}c_{2}-\rho_{1}c_{1}}{\rho_{2}c_{2}+\rho_{1}c_{1}}$$
where ρ and c denote the density of a medium and the sound speed in the medium, respectively. Equation (1) indicates that the main term affecting the reflection coefficient is ρc, which is commonly referred to as the characteristic impedance.

The acoustic impedance of a medium or material (Z) is defined as follows:(2)$$Z=\frac{P}{U}=X+iY$$
where P is the local acoustic pressure acting on the surface of the material and U is the local velocity of the fluid in the direction perpendicular to the material surface. The acoustic impedance is a complex number. The real and imaginary parts of the acoustic impedance are referred to as the acoustic resistance and acoustic reactance, respectively.

### 2.2. Sound Propagation in a Circular Tube

The propagation of one-dimensional sound waves in a circular tube obeys the wave equation, which can be expressed as follows:(3)$$\frac{\partial^{2}p}{\partial x^{2}}=\frac{1}{c^{2}}\frac{\partial^{2}p}{\partial t^{2}}$$
where p denotes the sound pressure, x is the axial coordinate of the circular tube, t is the time, and c is the sound velocity in a liquid. The parameter c can be expressed as follows: $c=\sqrt{E_{v}/\rho }$, where $E_{V}$ is the bulk modulus, and ρ is the density of the liquid. For a monochromatic sound wave, the following equation is obtained:(4)$$p(x,t)=P(x)e^{i\;\omega \;t}$$

When Equation (4) is substituted into Equation (3), the following equation is obtained:(5)$$\frac{d^{2}P}{dx^{2}}+k^{2}P=0$$

Equation (5) is known as the Helmholtz equation. The parameters ω and k in Equations (4) and (5), respectively, denote the angular frequency and wavenumber, respectively, and ω=k c.

Assume that the sound waves in a WFIT are plane waves; thus, the sound pressure field that satisfies Equation (5) can be expressed as follows:(6)$$P\left(x\right)=P_{i}e^{ik\left(L-x\right)}+P_{r}e^{-ik\left(L-x\right)}$$
where $i=\sqrt{-1}$ and $P_{i}e^{ik\left(L-x\right)}$ and $P_{r}e^{-ik\left(L-x\right)}$ represent the incident and reflected sound waves in the +x and −x directions, respectively. On the basis of Equation (6), the reflection coefficient R can be expressed as follows:(7)$$R\left(x\right)=\frac{P_{r}e^{-ik\left(L-x\right)}}{P_{i}e^{ik\left(L-x\right)}}$$

This study presented the complex reflection coefficient R in polar form; therefore, $R=\left|R\right|e^{i\beta }$, where $\left|R\right|$ and β are referred to as the amplitude (or modulus) and phase, respectively, of the reflection coefficient. The reflection coefficient at the surface of the material (when x=L) is expressed as follows:(8)$$R_{L}=\frac{P_{r}}{P_{i}}$$

Furthermore, the linearized form of the momentum equation for an inviscid fluid can be expressed as follows:(9)$$\rho \frac{\partial u}{\partial t}=-\frac{\partial p}{\partial x}$$
where u denotes the velocity of fluid particles. Assuming that sound waves are monochromatic, the following equation is satisfied:(10)$$u(x,t)=U(x)e^{i\;\omega \;t}$$

When Equations (2), (4) and (10) are substituted into Equation (9), the following equation is obtained:(11)$$P+\frac{Z_{L}}{\rho \left(i\omega \right)}\frac{\partial P}{\partial x}=0$$

By substituting Equation (6) into Equation (11) and setting x=L, one obtains the following equation:(12)$$P_{i}(1-\frac{Z_{L}}{\rho c})+P_{r}(1+\frac{Z_{L}}{\rho c})=0$$

Subsequently, the following equation can be derived:(13)$$R_{L}=\frac{P_{r}}{P_{i}}=\frac{Z_{L}/\rho c-1}{Z_{L}/\rho c+1}$$

Equation (13) can be rewritten as follows:(14)$$Z_{L}/\rho c=\frac{1+R_{L}}{1-R_{L}}$$

Equation (14) expresses the relationship between the reflection coefficient and the acoustic impedance.

### 2.3. Transfer Function Method

The transfer function method has been widely used in the measurement of the acoustic reflection coefficient of a material with an air impedance tube. A schematic for interpreting the transfer function method is displayed in Figure 2.

The transfer function H is defined as the complex ratio of the sound pressures $P_{1}\left(f\right)$ and $P_{2}\left(f\right)$ measured at the microphone locations A($x_{1}$) and B($x_{2}$), respectively.
(15)$$H=\frac{P_{2}\left(f\right)}{P_{1}\left(f\right)}$$

Parameters $P_{1}\left(f\right)$ and $P_{2}\left(f\right)$ can be expressed as follows:(16)$$\left\{\begin{array}{c} P_{1}=P_{i}e^{ik\left(L-x_{1}\right)}+P_{r}e^{-ik\left(L-x_{1}\right)}\\ P_{2}=P_{i}e^{ik\left(L-x_{2}\right)}+P_{r}e^{-ik\left(L-x_{2}\right)} \end{array}\right.$$

By substituting Equation (16) into Equation (15), the following equation is obtained:(17)$$H=\frac{P_{2}}{P_{1}}=\frac{P_{i}e^{ik\left(L-x_{2}\right)}+P_{r}e^{-ik\left(L-x_{2}\right)}}{P_{i}e^{ik\left(L-x_{1}\right)}+P_{r}e^{-ik\left(L-x_{1}\right)}}$$

On the basis of Equations (7) and (8), the relationship between the reflection coefficients at positions $x_{1}$ and x=L is determined as follows:(18)$$R_{x_{1}}=\frac{P_{r}e^{-ik\left(L-x_{1}\right)}}{P_{i}e^{ik\left(L-x_{1}\right)}}=R_{L}e^{-2ik\left(L-x_{1}\right)}$$

After Equation (18) is substituted into Equation (17), the following equation is obtained:(19)$$R_{x_{1}}=\frac{e^{-ik\left(x_{2}-x_{1}\right)}-H}{H-e^{ik\left(x_{2}-x_{1}\right)}}$$

Equation (19) can be substituted into Equation (18) to obtain the reflection coefficient $R_{L}$ at the material surface.
(20)$$R_{L}=\frac{e^{-ik\;s}-H}{H-e^{ik\;s}}e^{2ik(L-x_{1})}$$
where $s=x_{2}-x_{1}$ is the distance between microphones A and B. After the measurement of the transfer function H, the reflection coefficient of the material can be calculated using Equation (20). The reflection coefficient can be substituted into Equation (14) to determine the impedance value $Z_{L}$.

### 2.4. Sensor-Switching Technique

The transfer function method can easily be used to determine the reflection coefficient with an impedance tube; however, because the reflection coefficient is obtained from the ratio of sound pressures measured by two microphones, the amplitude and phase inconsistencies between the two microphones can cause errors in the transfer function. Chung and Blaser [5,6] proposed the sensor-switching technique to eliminate the differences in sensitivity between two sensors. Figure 3 presents a schematic for interpreting the sensor-switching technique.

Let $\alpha_{A}\left(f\right)$ and $\alpha_{B}\left(f\right)$ be the complex sensitivity of microphones *A* and *B*, respectively, and $V_{A}\left(f\right)$ and $V_{B}\left(f\right)$ be the voltage signals received by microphones *A* and *B*, respectively, where f is the sound frequency. The relationship between the received voltage signal and the sound pressure is expressed as follows: $V\left(f\right)=P\left(f\right)\cdot \alpha \left(f\right)$.

In standard measurements, as shown in the left part of Figure 3, the transfer function $H_{11}$ is defined as follows:(21)$$H_{11}=\frac{V_{B}}{V_{A}}=\frac{P_{2}\cdot \alpha_{B}}{P_{1}\cdot \alpha_{A}}$$

When the microphone positions are switched, as displayed in the right part of Figure 3, the transfer function $H_{12}$ can be obtained as follows:(22)$$H_{12}=\frac{V_{B}}{V_{A}}=\frac{P_{1}\cdot \alpha_{B}}{P_{2}\cdot \alpha_{A}}$$

Parameters $H_{11}$ and $H_{12}$ can be used to determine the modifier factor $H_{c}$ as follows:(23)$$H_{c}=\sqrt{H_{11}\times H_{12}}=\frac{\alpha_{B}}{\alpha_{A}}$$

By dividing Equation (21) by Equation (23), one can eliminate the influence of sensitivity differences between microphones to obtain the actual sound pressure transfer function H.
(24)$$H=\frac{H_{11}}{H_{c}}=\frac{P_{2}}{P_{1}}$$

Parameters $H_{11}$ and $H_{12}$ are obtained through measurements (Figure 3). Subsequently, $H_{c}$ is determined using Equation (23), and the transfer function H is obtained from Equation (24). The value of H can be substituted into Equation (20) to obtain the reflection coefficient.

### 2.5. Three-Parameter Calibration Method

The two-microphone 3PCM is used to correct many measurement errors, including circuit errors between receiving systems, sensitivity errors between two hydrophones, the amplitude attenuation error when sound waves are transmitted in water, and the phase variation error.

As in the case of the sensor-switching technique, let $\alpha_{A}\left(f\right)$ and $\alpha_{B}\left(f\right)$ be the complex sensitivity of hydrophones *A* and *B*, respectively, and let $V_{A}\left(f\right)$ and $V_{B}\left(f\right)$ be the voltage signals received by hydrophones *A* and *B*, respectively. Assume that the relationship between the voltage signal and the sound pressure can be expressed as follows: $V\left(f\right)=P\left(f\right)\cdot \alpha \left(f\right)$. In this case, the transfer function expressed in Equation (15) can be rewritten as follows:(25)$$H=\frac{V_{B}}{V_{A}}=\frac{P_{b}\cdot \alpha_{B}}{P_{a}\cdot \alpha_{A}}$$

The sound pressures measured by two hydrophones can be expressed as follows:(26)$$\left\{\begin{array}{c} P_{a}=P_{i}\alpha_{i1}+P_{r}\alpha_{r1}\\ P_{b}=P_{i}\alpha_{i2}+P_{r}\alpha_{r2} \end{array}\right.$$
where $\alpha_{i1}$, $\alpha_{r1}$, $\alpha_{i2}$ and $\alpha_{r2}$ represent complex calibration factors for the modulus and phase of the pressure. By substituting Equation (26) into Equation (25), the following equation is obtained:(27)$$H=\frac{\left(P_{i}\alpha_{i2}+P_{r}\alpha_{r2}\right)\cdot \alpha_{B}}{\left(P_{i}\alpha_{i1}+P_{r}\alpha_{r1}\right)\cdot \alpha_{A}}$$

By substituting the reflection coefficient obtained using Equation (13) into Equation (27), one obtains the following equation:(28)$$R=\frac{\lambda_{1}H+\lambda_{2}}{H+\lambda_{3}}$$

The parameters $\lambda_{1}$, $\lambda_{2}$, and $\lambda_{3}$, which are unknown, can be determined as follows:(29)$$\lambda_{1}=-\frac{\alpha_{i1}}{\alpha_{r1}},\;\lambda_{2}=\frac{\alpha_{i2}\alpha_{B}}{\alpha_{r1}\alpha_{A}},\;\lambda_{3}=-\frac{\alpha_{r2}\alpha_{B}}{\alpha_{r1}\alpha_{A}}$$

Assume that the reflection coefficients of three reference materials are known and denoted as $R_{1}$, $R_{2}$, and $R_{3}$. Moreover, assume that the corresponding transfer functions obtained from measurements in an impedance tube for the three materials are $H_{1}=V_{B}^{\left(1\right)}/V_{A}^{\left(1\right)}$, $H_{2}=V_{B}^{\left(2\right)}/V_{A}^{\left(2\right)}$, and $H_{3}=V_{B}^{\left(3\right)}/V_{A}^{\left(3\right)}$. After the aforementioned values are substituted into Equation (28), $\lambda_{1}$, $\lambda_{2}$, and $\lambda_{3}$ can be determined as follows:(30)$$\left\{\begin{array}{c} \lambda_{1}=-\frac{R_{2}R_{3}(H_{3}-H_{2})+R_{1}R_{3}(H_{1}-H_{3})+R_{1}R_{2}(H_{2}-H_{1})}{R_{1}(H_{3}-H_{2})+R_{2}(H_{1}-H_{3})+R_{3}(H_{2}-H_{1})}\\ \lambda_{2}=\frac{R_{2}R_{3}H_{1}(H_{3}-H_{2})+R_{1}R_{3}H_{2}(H_{1}-H_{3})+R_{1}R_{2}H_{3}(H_{2}-H_{1})}{R_{1}(H_{3}-H_{2})+R_{2}(H_{1}-H_{3})+R_{3}(H_{2}-H_{1})}\\ \lambda_{3}=\frac{R_{1}H_{1}(H_{3}-H_{2})+R_{2}H_{2}(H_{1}-H_{3})+R_{3}H_{3}(H_{2}-H_{1})}{R_{1}(H_{3}-H_{2})+R_{2}(H_{1}-H_{3})+R_{3}(H_{2}-H_{1})} \end{array}\right.$$

To determine $\lambda_{1}$, $\lambda_{2}$, and $\lambda_{3}$, theoretically, the reflection coefficients of three reference materials must be known, which is practically impossible. When sound waves are incident on a water–air interface, negative total reflection occurs with a phase difference of 180° (i.e., $R_{L}=R_{water/air}=e^{i\pi }$) because of the large difference in the characteristic impedance between these two media (Equation (1)). This characteristic can be used to determine the aforementioned three unknown parameters. Three water columns of different heights with a free surface can be used to determine these parameters.

As displayed in Figure 4, three water columns with different column heights $d_{i}$ (i=1, 2, 3) can be selected by appropriately stretching the same measurement surface to the free surface. The reflection coefficient at the measurement surface $R_{i}$ can be determined from Equation (18) as follows:(31)$$R_{i}=R_{water/air}\cdot e^{-2ikd_{i}}=e^{i\left(\pi -2kd_{i}\right)}$$

The three calibration parameters $\lambda_{1}$, $\lambda_{2}$, and $\lambda_{3}$ can be determined by substituting the theoretical reflection coefficients ($R_{1}$, $R_{2}$, and $R_{3}$) obtained from Equation (31) and the transfer functions ($H_{1}$, $H_{2}$, and $H_{3}$) measured using hydrophones into Equation (30). If a sound-absorbing material is placed at a distance of $d^{*}$ from the measurement surface, the material’s reflection coefficient $R^{**}$ can be determined using Equation (28) after measuring the transfer function H. By substituting $R^{**}$ into Equation (18), the actual reflection coefficient at the surface of the sound-absorbing material can be determined as follows:(32)$$R^{*}=R^{**}\cdot e^{2ikd^{*}}$$

## 3. Instrumentation of a WFIT

### 3.1. Design

No standard method exists for measuring the impedance and absorption of an acoustical material using a WFIT. This study followed the design specifications of the air acoustic impedance tube described in ASTM E1050-12 [2] to develop a pressurized WFIT.

The operating frequency of this impedance tube is defined as follows:(33)$$f_{\ell }<f<f_{u}$$
where f, $f_{\ell }$, and $f_{u}$ represent the operating frequency, lower operating frequency, and upper operating frequency of the tube, respectively. In an impedance tube, the upper limit of the operating frequency is expressed as follows:(34)$$f_{u}<\frac{Kc}{b}$$
where K=0.586, c is the sound velocity in the tube, and b is the inner diameter of the tube. The lower operating frequency limit $f_{\ell }$ should satisfy the following equation:(35)$$f_{\ell }>(c/s)\times 1\%$$
where s is the hydrophone spacing.

At a location close to the sound transmitter, the emitted sounds may not fully develop to become plane waves. Therefore, hydrophones should be mounted a long distance from the transmitter. The distance between the first hydrophone and the transmitter is suggested to be at least three times the inner diameter of the tube.

Furthermore, when the tube is filled with water rather than air, because the characteristic impedance of the water and tube wall (usually stainless steel) are close to each other, the tube wall can no longer be treated as a rigid body. In the aforementioned situation, elastic waves of the tube wall interfere with sound waves in the water, and the assumption that the sound waves are plane waves is no longer valid. According to the elastic waveguide theory proposed by Del Grosso [11], when the medium in an impedance tube is water, the assumption of plane wave propagation is invalid. However, when the medium in an impedance tube is steel and the wall thickness is large, the sound waves of the lowest-order propagation mode in the tube approximate plane waves [8]. Jian [14] determined the sound velocity $c_{eff}$ in this mode to be 1458 m/s, which is close to the sound velocity in water at 20 °C (i.e., 1480 m/s).

Based on the aforementioned discussion, 304L stainless steel was used to construct a tube with a length of 150 cm, an outer diameter of 17.6 cm, and an inner diameter of 11.6 cm. The hydrophone spacing s was set as 6.5 cm. Based on Equations (34) and (35) and $c=c_{eff}=1458\;m/s$, $f_{u}$ and $f_{\ell }$ were determined as 7365 and 224 Hz, respectively. Table 1 presents the material and wave parameters of the developed pressurized WFIT.

### 3.2. Pressurized WFIT

A closed pressurized WFIT (Figure 5) was manufactured to study the acoustic properties of materials under various hydrostatic pressures. A compressor (water pump) was used to provide the required pressure for the conducted experiments. An upper tube with a length of 16 cm was installed on the top of the impedance tube to fix the test sample (Figure 6), and a steel backing plate was used to create a closed system (Figure 5 and Figure 7). This plate was equipped with a pressure gauge to indicate the pressure value inside the tube. A venting valve at the top of the tube was used to release bubbles from the liquid in the tube during the compression process. This valve was also used to release the pressure in the tube after the experiment.

In the experiments conducted to determine the acoustic properties in the pressurized WFIT, the compressor was turned off when the pressure inside the tube reached the desired value. Measurements started after the compressor was turned off for 15 min to ensure that the pressure inside the tube was stable and that water did not leak out from the tube.

### 3.3. Experimental Setup

The experimental setup used in this study to measure the reflection coefficients of an acoustic material under various hydrostatic pressures with the developed WFIT comprised of two parts: a sound-emitting system and sound-receiving system (Figure 8). The sound-emitting system that was adopted, which works within the operating frequency range of the NEPTUNE-TX335 transducer, is controlled using LabVIEW software. This system transmits signals through an analog-to-digital (A/D) converter (NI PCI-6110, NI BNC-2110) to a power amplifier, which drives the transducer to emit acoustic vibrations with a frequency of 2–7 kHz. This frequency range lies within the upper ($f_{u}=7365\;\mathrm{Hz}$) and lower ($f_{\ell }=224\;\mathrm{Hz}$) operating frequency limits specified in Section 3.1. The generated sound pressure was sampled at a rate of 50 kHz by two hydrophones (Brüel and Kjær Type 8104, Brüel and Kjær, Virum, Denmark) mounted on different positions of the impedance tube. The received signals were amplified by a charge amplifier (Brüel and Kjær NEXUS-2692), converted to digital signals by an A/D converter, and stored in a computer for subsequent data analysis. The sounds were emitted with a period of 0.21 s and a signal duration of 0.15 s. The LabVIEW parameters of the sound-emitting system are presented in Table 2.

### 3.4. Measurement Procedure

The first step in the measurement procedure involved filling the impedance tube with water at least 24 h before its operation. This step ensured temperature equilibrium in the tube, the release of entrained air bubbles from the tube, and the dissolution of entrapped air in the tube. However, residual bubbles possibly existed in the tube even after the aforementioned step. To avoid measurement uncertainties caused by the presence of bubbles, Wilson et al. [8] filled an impedance tube with distilled and degassed water several hours before conducting their experiment.

The 3PCM was used to calibrate the developed WFIT, and three water columns of different heights with a free surface on the top and the same measurement surface at the bottom were selected to determine the values of $\lambda_{i}\;(i=1,2,3)$. A Vernier caliper was used to measure the distance between the water surface of different water columns and the top of the tube. At least three values were measured and averaged for the aforementioned distance. The heights of the water columns were determined from the average distance between the water surface and the top of the tube, as well as the position of the measurement surface.

After the height of the water column $d_{i}\;(i=1,2,3)$ was obtained, the corresponding reflection coefficient $R_{i}\;(i=1,2,3)$ at the measurement surface was determined using Equation (31). The transfer functions $H_{i}\;(i=1,2,3)$ were then measured and substituted into Equation (30) to determine the values of $\lambda_{i}\;(i=1,2,3)$. The values of $\lambda_{i}\;(i=1,2,3)$ were substituted into Equation (28) to determine the reflection coefficient of the acoustic material installed in the impedance tube.

## 4. Results and Discussion

### 4.1. Reflection Coefficient at the Water–Air Interface

The developed WFIT was used to measure acoustic reflection coefficients at the water–air interface in accordance with the 3PCM, and the repeatability and accuracy of the experimental results were examined. The aforementioned reflection coefficients were compared with those obtained in accordance with the transfer function method and sensor-switching technique. In the experiments described in this section, the upper tube for fixing the test sample and the steel cover were not mounted on the WFIT, and measurements were conducted at frequency intervals of 10 Hz.

Figure 9a,b presents the amplitude and phase, respectively, of the reflection coefficients at the water–air interface obtained using the 3PCM and by repeating the measurement procedure described in Section 3.4 thrice. The sound frequency ranged from 2 to 7 kHz. The aforementioned results revealed high repeatability in the measurements, with a deviation of ±0.03 in the amplitude and a deviation of ±2° in the phase.

Figure 10a,b illustrates the averaged values of the amplitudes and phases presented in Figure 9a,b, respectively. As indicated in Section 2.3, the acoustic reflection coefficient at the water–air interface ($e^{i\pi }$) had an amplitude of 1.0 and a phase of 180°. A comparison of the experimental results displayed in Figure 10a,b with corresponding theoretical values revealed that the measurement errors in the amplitude and phase were approximately 0.02 and 4°, respectively. The maximum amplitude error occurred at f = 6570 Hz. The phase errors increased with the sound frequency. A possible reason for this result is as follows: to determine the values of $\lambda_{i}(i=1,\;2,\;3)$, the reflection coefficients $R_{i}(i=1,2,3)$ at the measurement surface must be provided, which are equal to $e^{i\;(\pi -2kd_{i})}$ with k=2πf/c; hence, the errors that occurred in the measurement of $d_{i}$ will result in frequency-dependent phase errors in the $R_{i}$ values. Subsequently, these errors in the $R_{i}$ values resulted in the frequency-dependent phase errors in the reflection coefficients.

Figure 11a,b displays the amplitudes and phases, respectively, of the reflection coefficients at the water–air interface that were obtained using the 3PCM, transfer function method, and sensor-switching technique. Among the aforementioned results, the reflection coefficients obtained using the 3PCM were the closest to the theoretical values, followed by those obtained using the sensor-switching technique and transfer function method.

Table 3 lists the maximum errors obtained between theoretical and measured reflection coefficients at the water–air interface in various previous studies [7,8,14] and the present study. The data in Table 3 reveal that the error in the measured reflection coefficient was considerably smaller when the 3PCM was used than when the sensor-switching technique was used. The accuracy of the reflection coefficient measured by Wilson et al. [8] by using the 3PCM was higher than that measured using the 3PCM in this study. Some possible reasons for this result are presented as follows. First, Wilson et al. [8] filled their tube with distilled and degassed water, while, in this study, tap water was used. Second, Wilson et al. [8] designed an elaborate apparatus to substantially reduce the disturbance of hydrophones in the sound field.

### 4.2. Repeatability of Reflection Coefficient Measurements for Porous Rubber

The present study and the study of Wilson et al. [8] indicate that high repeatability and accuracy can be achieved using measurements of the acoustic reflection coefficient at the water–air interface when using the 3PCM with a WFIT. However, few studies [17] have applied this method to obtain the reflection coefficient of an acoustic material in a WFIT.

In this section, the acoustic reflection coefficient of porous rubber was measured using the 3PCM with the developed WFIT. The adopted porous rubber specimen had a diameter of 11.52 cm and a thickness of 6.53 cm and was placed in the tube with its back side at the same level as the top of the tube. As the specimen was porous, the specimen was immersed in water for at least 4 h before each experiment to ensure that all its pores were fully saturated with water. Figure 12 displays an image of the porous rubber specimen. Calibration tests were performed to obtain the values of $\lambda_{1}$, $\lambda_{2}$, and $\lambda_{3}$. Table 4 lists the values of $\lambda_{i}(i=1,\;2,\;3)$ obtained in three tests conducted on different dates.

In the three aforementioned tests, the entire measurement procedure described in Section 3.4 was applied—from filling the tube with water to measuring the transfer function. The heights of water columns $d_{i}(i=1,2,3)$ were set to be identical in each test. For simplicity, $d_{1}$, $d_{2}$, and $d_{3}$ were set as 10, 20, and 30 mm, respectively.

To ensure that the determined values of $\lambda_{i}(i=1,\;2,\;3)$ and the measurements conducted with the WFIT were accurate, a fourth water column was set with a height of 40 mm from the measurement surface, and the $\lambda_{i}(i=1,\;2,\;3)$ values were used to determine the reflection coefficient at the water–air interface of this column using Equation (28). The measurement surface for the fourth column was set such that the distance from the free surface of this column to the top of the tube was equal to the thickness of the test specimen. Thus, the back side of the specimen was at the same level as the top of the tube.

Figure 13a,b depicts the amplitudes and phases, respectively, of the measured reflection coefficients of the porous rubber specimen that were obtained in three tests by using the developed WFIT. Although measurements were taken at a 10-Hz interval, the phase values in Figure 13b are shown with a 100-Hz interval for clarity. The results in Figure 13 indicate that the amplitudes and phases of the acoustic reflection coefficients of the porous rubber specimen did not vary considerably between the three tests. The results in the three tests were not completely identical to each other possibly because the physical conditions—such as the heights of the water columns, the sample position, and the bubble size and number—might have varied marginally between the three tests. As a phase angle of 180° is identical to that of −180°, the results in Figure 13b indicate that the phase angle of the reflection coefficient at the surface of the porous rubber specimen (approximately 180°) was very close to that at the water–air interface; see Figure 10b.

### 4.3. Reflection Coefficients Obtained for Porous Rubber Specimens under Different Hydrostatic Pressures

Additional porous rubber specimens were used to demonstrate the ability of the developed pressurized WFIT to measure acoustic reflection coefficients under various hydrostatic pressures. The 3PCM was applied to determine the reflection coefficients. The calibration procedure for determining the three calibration parameters was the same as that described in the previous section. The specimens were placed in the upper tube designed for fixing the test sample (Figure 6). The back side of the specimens had a steel cover (refer to Figure 5 and Figure 7). The thickness of each layer of the specimens was 25 mm. Specimen layers were stuck together using adhesive to obtain specimens with different thicknesses (Figure 14). As the specimens were porous, they were immersed in water for at least 4 h before each experiment to ensure that all their pores were fully saturated with water.

Figure 15 presents the amplitudes of the reflection coefficients of a single-layer rubber specimen under a hydrostatic pressure of 4 P_atm_ when measurements were taken thrice on the same day using the same experimental setup. Figure 16 presents the amplitudes of the reflection coefficients of the single-layer rubber specimen under a hydrostatic pressure of 4 P_atm_ when repeating the measurement procedure described in Section 3.4 on three dates. The repeatability of the results displayed in Figure 15 is high. The repeatability of the results displayed in Figure 16 is acceptable but lower than that of the results displayed in Figure 15. The aforementioned phenomenon possibly occurs because the physical conditions marginally varied between the tests conducted on the same day and those conducted on different days, which possibly resulted in minor deviations in the reflection coefficients. The phase results corresponding to the amplitude results displayed in Figure 15 and Figure 16 are not shown in this paper because, as indicated by Figure 13, when the amplitudes of reflection coefficients are close to each other, their phase values are also close to each other. In the study of Fu et al. [18], the repeatability of sound-absorbing coefficient obtained in a pressurized WFIT using the transfer function method was unsatisfactory, with the maximum deviation reaching 20% at some frequencies.

After verifying the repeatability of the experimental results, the acoustic properties of porous rubber specimens with different thicknesses (25, 50, and 75 mm) were systematically investigated under various hydrostatic pressures (1, 4, and 8 P_atm_). Table 5 presents the conditions in this investigation. The calibration procedure was performed for each specimen to obtain corresponding values for the three calibration parameters.

Figure 17a,b displays the amplitudes and phases of the measured reflection coefficients for a specimen with a thickness of 25 mm (Case 1). In Case 1, when the hydrostatic pressure increased from 1 to 4 P_atm_, the reflection coefficient increased in the frequency range of 2–4 kHz, with the lowest reflection coefficient being approximately 0.5 in this frequency range; however, in the frequency range of 4–7 kHz, the reflection coefficient substantially decreased, with a minimum value of approximately 0.45 at a frequency of approximately 6700 Hz. An increase in the hydrostatic pressure from 4 to 8 P_atm_ resulted in a decrease in the reflection coefficient in the frequency range of 2–4 kHz and an increase in the reflection coefficient in the frequency range of 4–7 kHz. Figure 17b reveals that pressure variations did not result in notable phase changes. The phase values were approximately 180°, which was close to the value obtained under an unpressurized condition (Section 4.2).

Figure 18a,b presents the amplitudes and phases, respectively, of the measured reflection coefficients for a 50-mm-thick specimen (Case 2). In Case 2, when the hydrostatic pressure increased from 1 to 4 P_atm_, the reflection coefficient substantially decreased in the frequency ranges of 2.5–4 and 6.0–6.7 kHz, with the lowest reflection coefficient of 0.6 at approximately 3.2 and 6.7 kHz. A further increase in the hydrostatic pressure to 8 P_atm_ resulted in a decrease in the reflection coefficient compared with the values obtained under 1 P_atm_. As displayed in Figure 18b, the phase of the reflection coefficient did not change considerably with increases in the hydrostatic pressure.

Figure 19a,b displays the amplitudes and phases, respectively, of the reflection coefficients obtained for a 75-mm-thick specimen (Case 3). In Case 3, when the hydrostatic pressure increased from 1 to 4 P_atm_, the reflection coefficient did not exhibit a consistent increasing or decreasing trend in the frequency range of 2.0–4.0 kHz but remained approximately constant in the frequency range of 4–6.5 kHz. A further increase in the hydrostatic pressure to 8 P_atm_ resulted in a decrease in the reflection coefficient. As displayed in Figure 19b, the phase of the reflection coefficient did not considerably change with increases in the hydrostatic pressure.

The results displayed in Figure 17a and Figure 18a reveal that, under the studied hydrostatic pressures, the reflection coefficients in Cases 1 and 2 mainly ranged from 0.6 to 0.9; however, in Case 3 (Figure 19a), the reflection coefficient ranged from 0.50 to 0.78. The aforementioned results indicate that an increase in the specimen thickness from 25 to 50 mm did not result in notable changes in the reflection coefficient; however, an increase in the specimen thickness to 75 mm resulted in a decrease in the reflection coefficient in the studied frequency range.

The results illustrated in Figure 17, Figure 18 and Figure 19 indicated that in the investigated porous rubber specimens, the reflection coefficient was not obviously correlated with hydrostatic pressure. An increase in the hydrostatic pressure might not lead to an increase in the reflection coefficient in the studied sound-frequency range. This phenomenon might be related to the pore sizes and pore structure of the specimens, or the adhesive used to stack multiple layers to produce multilayer specimens. The pore sizes and pore structure of the specimens were unknown to the authors of this study.

The mechanisms that cause underwater sound attenuation in porous rubber might include wave redirection at the specimen surface, scattering by pores or inhomogeneities, conversion of the wave propagation mode at specimen boundaries, and the intrinsic absorption of sound waves by viscous pore fluids or viscoelastic materials through the conversion of these waves into heat. Thus, the pore size and pore structure play crucial roles in achieving satisfactory sound-absorbing performance. Determining the relationships between pore size, pore structure, and sound-absorbing performance can aid in the manufacturing of porous materials with a high sound-absorbing performance; however, the investigation of these relationships is beyond the scope of the present study. Additional information on underwater sound-absorbing materials can be obtained from the studies of Jayakumari et al. [28], Guillermic et al. [29], Dong and Tian [30], and Fu et al. [18].

Furthermore, under a pressurized condition, the speed of sound increases with an increase in pressure values. According to Wilson [31], the speed of sound in distilled water with a temperature of 20 °C and a pressure of 1 P_atm_ is 1482.92 m/s and that an increase in pressure to 4 and 8 P_atm_ causes the corresponding speed of sound to increase to 1483.42 and 1484.09 m/s, respectively. Notably, Equation (20) indicates that an increase in the speed of sound, or a decrease in the wavenumber, affects the reflection coefficient. However, in the studied pressure range, the increase in the speed of sound caused by an increase in pressure is minuscule. Therefore, the effect of an increase in the speed of sound on the reflection coefficient is negligible.

## 5. Conclusions

On the basis of the standard test method described in ASTM E1050-12 [2] for an air-filled impedance tube, a vertical pressurized WFIT was developed in this study to investigate the sound-absorbing characteristics of acoustic materials under various hydrostatic pressures. The 3PCM was employed with the developed WFIT to measure the reflection coefficients of materials. Mono-frequency sounds with a frequency of 2 to 7 kHz were adopted as incident signals, and measurements were conducted with a 10-Hz interval. The relationship between the reflection coefficient and the acoustic impedance on the material surface was determined.

First, the developed WFIT was used to measure the reflection coefficient at its water–air interface. The reflection coefficients obtained using the 3PCM were closer to the theoretical values than those obtained using the transfer function method and sensor-switching technique. The maximum measurement errors in the amplitude and phase of the reflection coefficient at the water–air interface were 0.02 and 4°, respectively, when using the 3PCM.

Second, the reflection coefficient of a porous rubber specimen with a diameter of 11.52 cm and a thickness of 6.53 cm was measured using the developed WFIT and the 3PCM. The reflection coefficients obtained in three tests conducted on this specimen on different dates exhibited high repeatability.

Third, the developed pressurized WFIT was used to obtain the reflection coefficient of a porous rubber specimen with a thickness of 25 mm under a hydrostatic pressure of 4 P_atm_. The reflection coefficient was determined by conducting measurements three times on the same day using the same experimental setup and one time each over three days by repeating the same measurement procedure. The results obtained in these tests exhibited high repeatability, which demonstrated the good performance of the developed WFIT and 3PCM to measure the acoustic reflection coefficient of sound-absorbing materials.

Finally, the reflection coefficients of porous rubber specimens with thicknesses of 25, 50, and 75 mm were systematically measured under hydrostatic pressures of 1, 4, and 8 P_atm_ to study their sound-absorbing behaviors. The experimental results revealed no obvious correlation between the reflection coefficient and the hydrostatic pressure for these specimens. For the aforementioned specimens, an increase in the hydrostatic pressure did not necessarily lead to an increase in the reflection coefficient in the studied sound frequency range. Furthermore, with an increase in the specimen thickness from 25 to 75 mm, the reflection coefficient decreased in the studied frequency range.

## Figures and Tables

**Figure 1 sensors-22-03827-f001:**
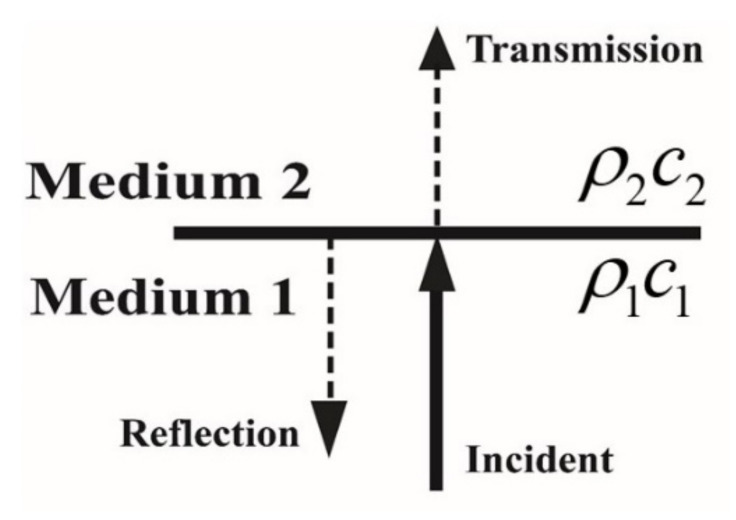
Sound transmission at the interface between two media.

**Figure 2 sensors-22-03827-f002:**
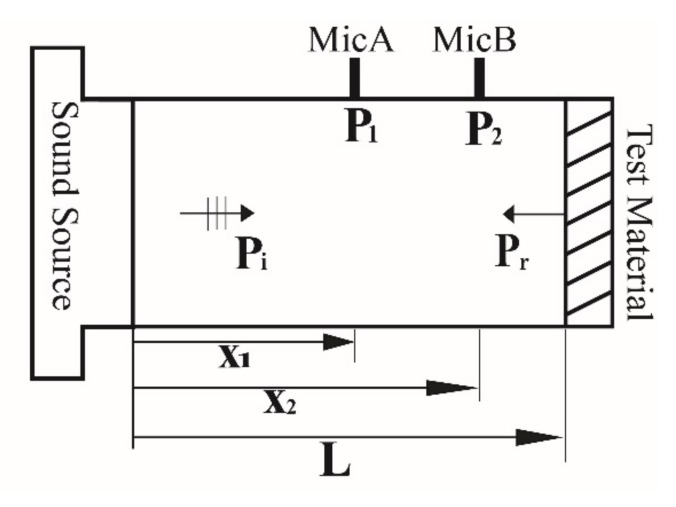
Schematic for interpreting the transfer function method.

**Figure 3 sensors-22-03827-f003:**
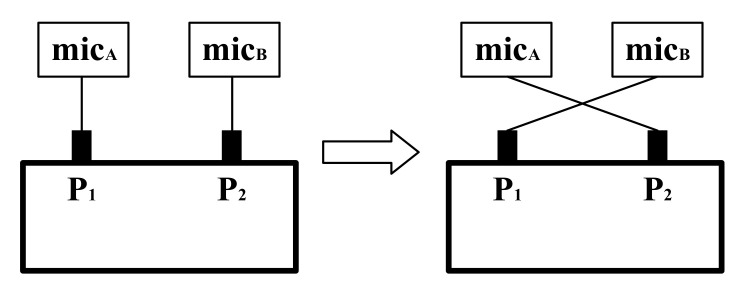
Schematic for interpreting the sensor-switching technique.

**Figure 4 sensors-22-03827-f004:**
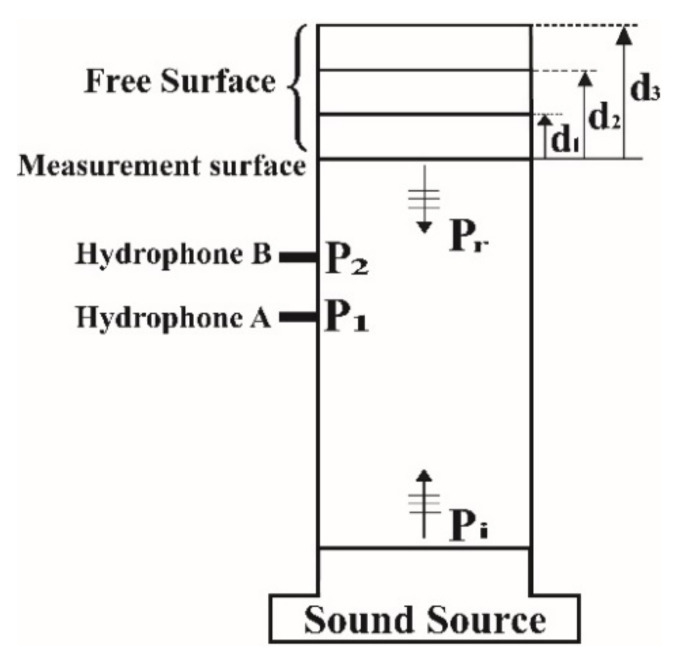
Schematic of a vertical water-filled impedance tube (WFIT).

**Figure 5 sensors-22-03827-f005:**
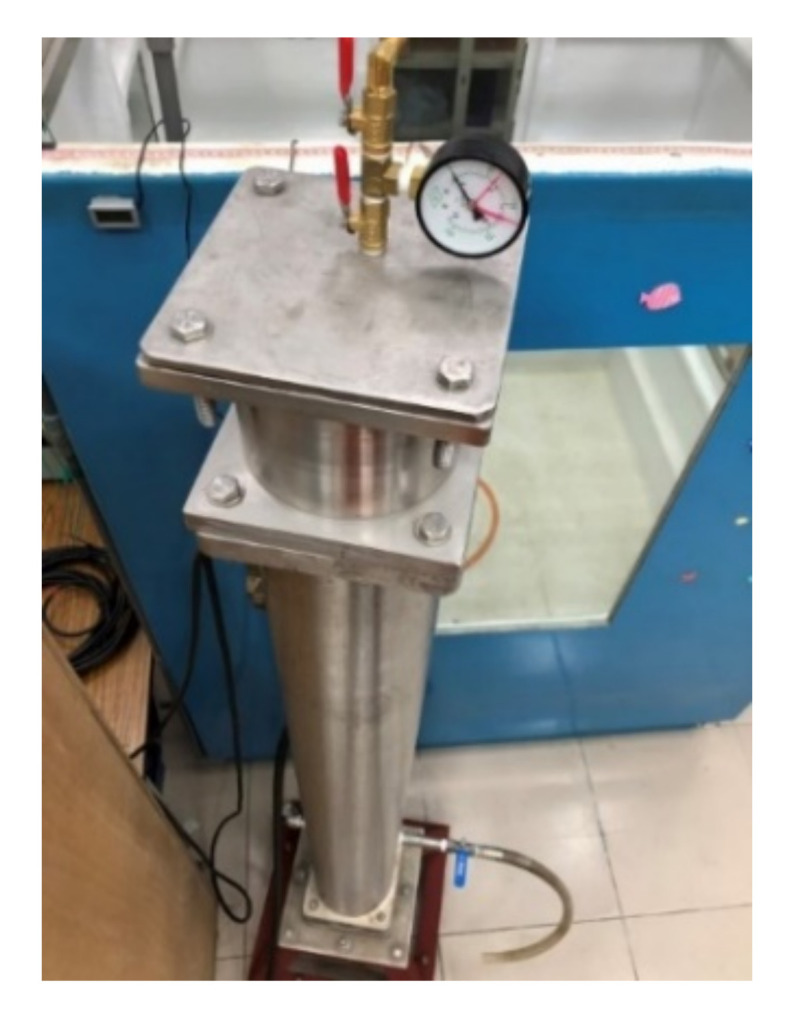
Appearance of the pressurized WFIT.

**Figure 6 sensors-22-03827-f006:**
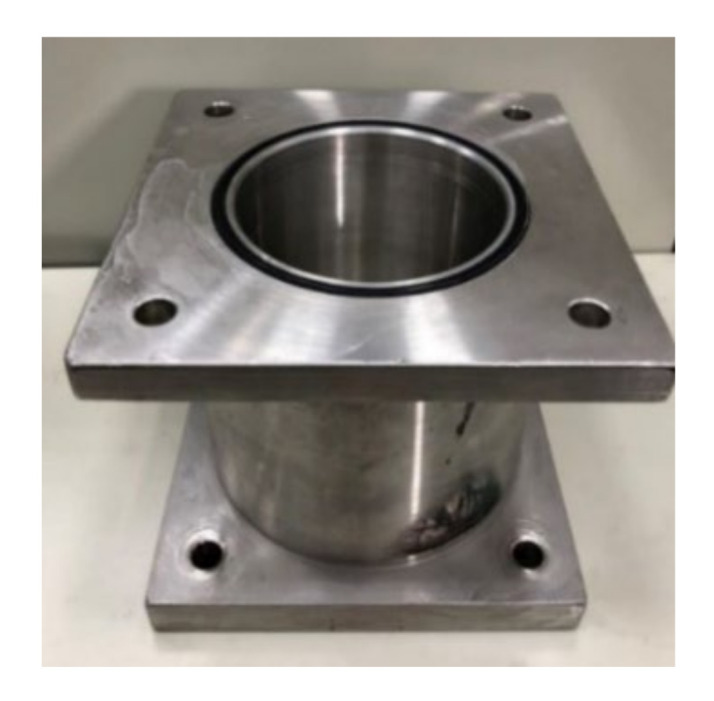
Upper tube for fixing specimen.

**Figure 7 sensors-22-03827-f007:**
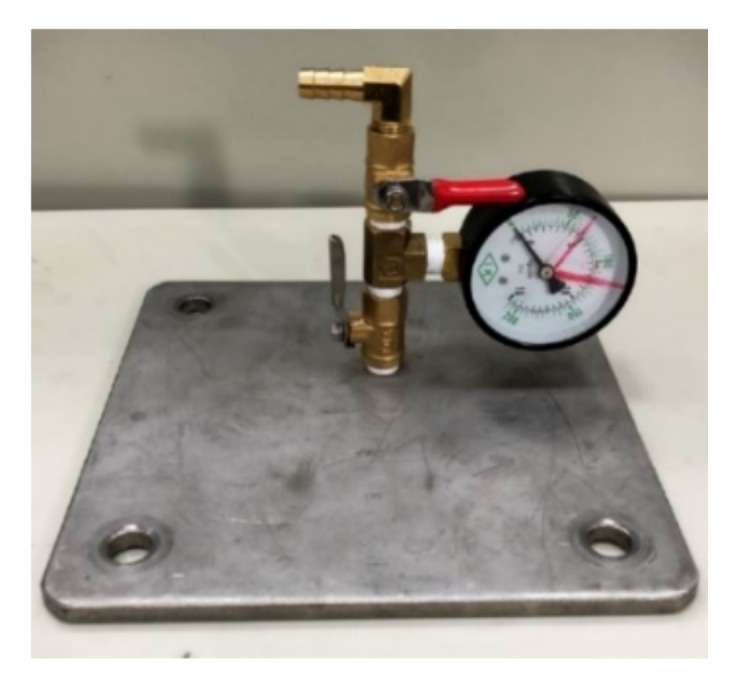
Seal cover of the pressurized WFIT.

**Figure 8 sensors-22-03827-f008:**
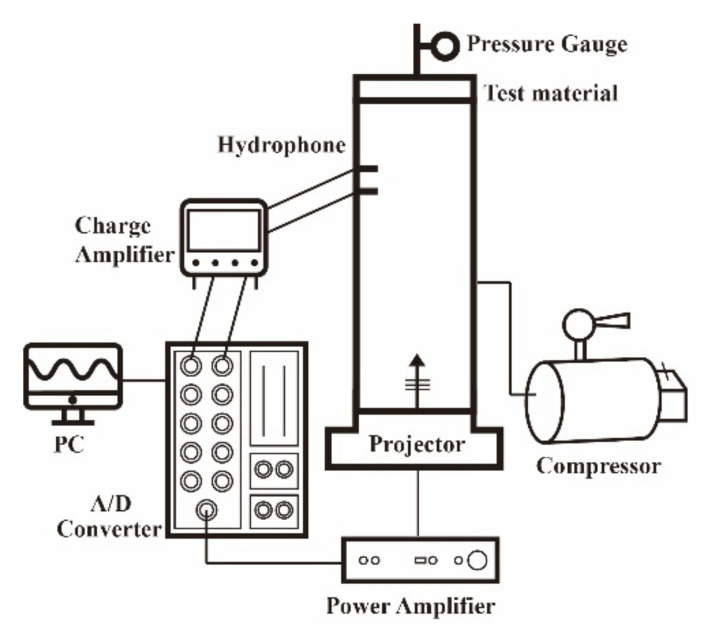
Experimental setup to measure the reflection coefficients of acoustic materials under various hydrostatic pressures using the developed pressurized WFIT.

**Figure 9 sensors-22-03827-f009:**
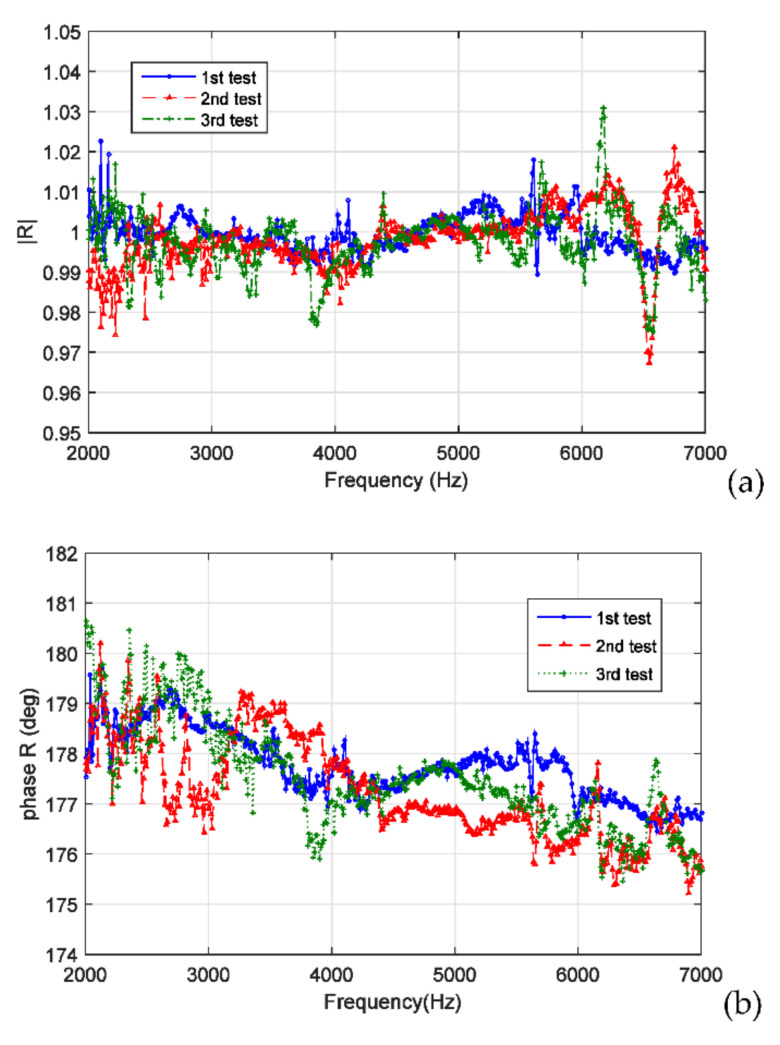
(**a**) Amplitude and (**b**) phase of the acoustic reflection coefficients at the water–air interface when the two-microphone, three-parameter calibration method (3PCM) was used and when the same measurement procedure was repeated thrice.

**Figure 10 sensors-22-03827-f010:**
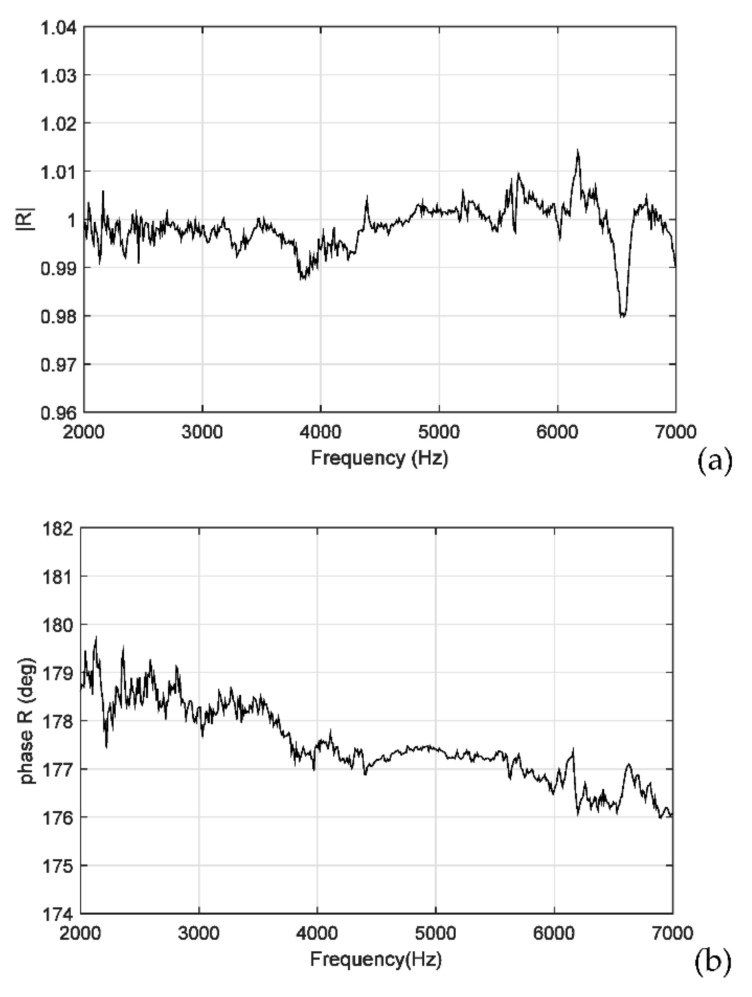
Averaged values of the (**a**) amplitudes and (**b**) phases presented in Figure 9.

**Figure 11 sensors-22-03827-f011:**
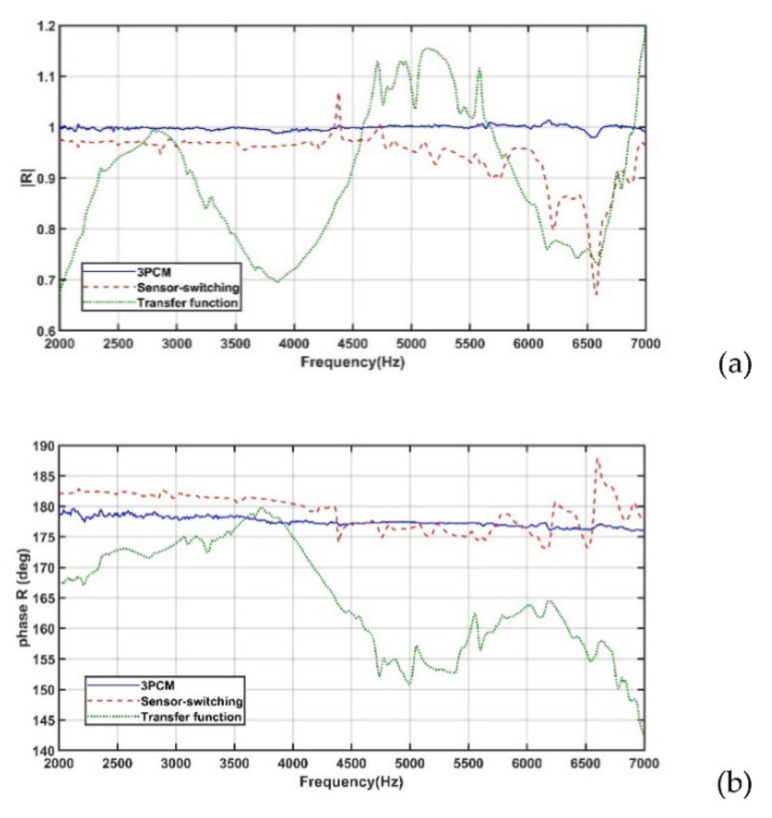
(**a**) Amplitudes and (**b**) phases of the measured acoustic reflection coefficients at the water–air interface that were obtained using the transfer function method, sensor-switching technique, and 3PCM.

**Figure 12 sensors-22-03827-f012:**
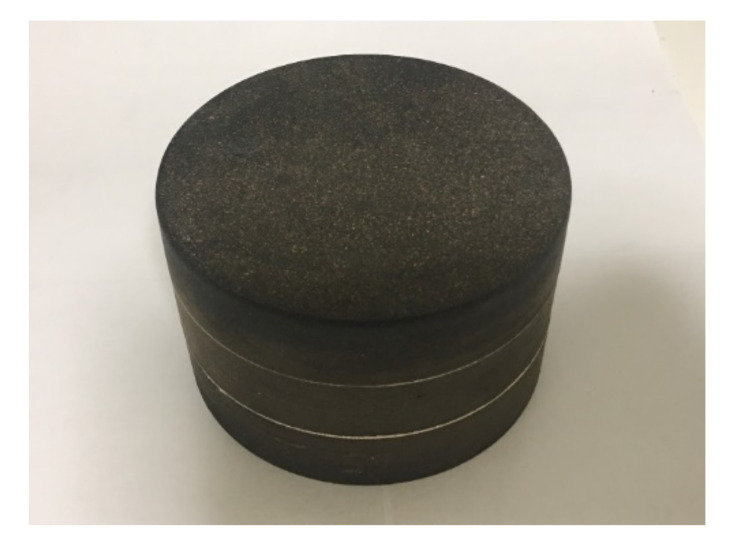
Image of a porous rubber specimen with a diameter of 11.52 cm and a thickness of 6.53 cm.

**Figure 13 sensors-22-03827-f013:**
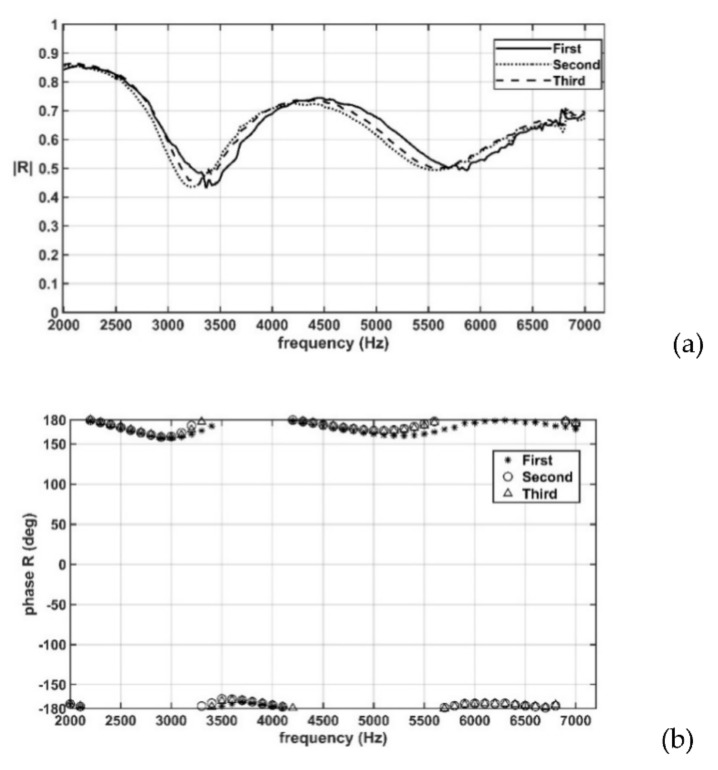
(**a**) Amplitudes and (**b**) phases of the measured acoustic reflection coefficients of a porous rubber specimen in the developed WFIT that were obtained in three tests conducted on different dates.

**Figure 14 sensors-22-03827-f014:**
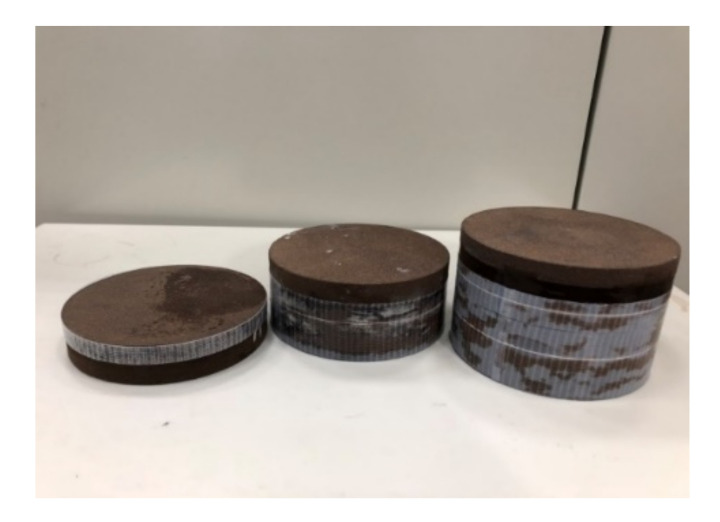
Porous rubber specimens of different thicknesses.

**Figure 15 sensors-22-03827-f015:**
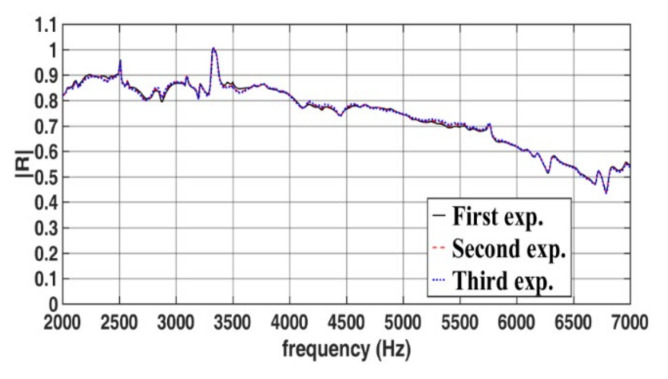
Amplitudes of the reflection coefficients of the single-layer specimen in the developed pressurized WFIT under a hydrostatic pressure of 4 P_atm_ when measurements were taken thrice on the same day using the same experimental setup.

**Figure 16 sensors-22-03827-f016:**
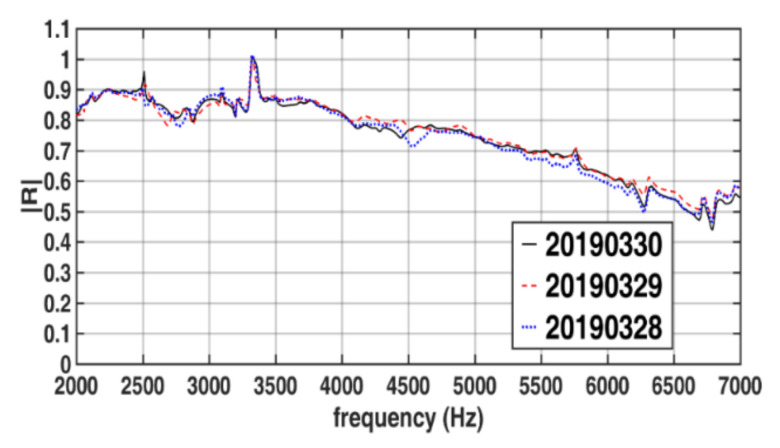
Amplitudes of the reflection coefficients of the single-layer specimen in the developed pressurized WFIT under a hydrostatic pressure of 4 P_atm_ when the same measurement procedure was applied on three dates.

**Figure 17 sensors-22-03827-f017:**
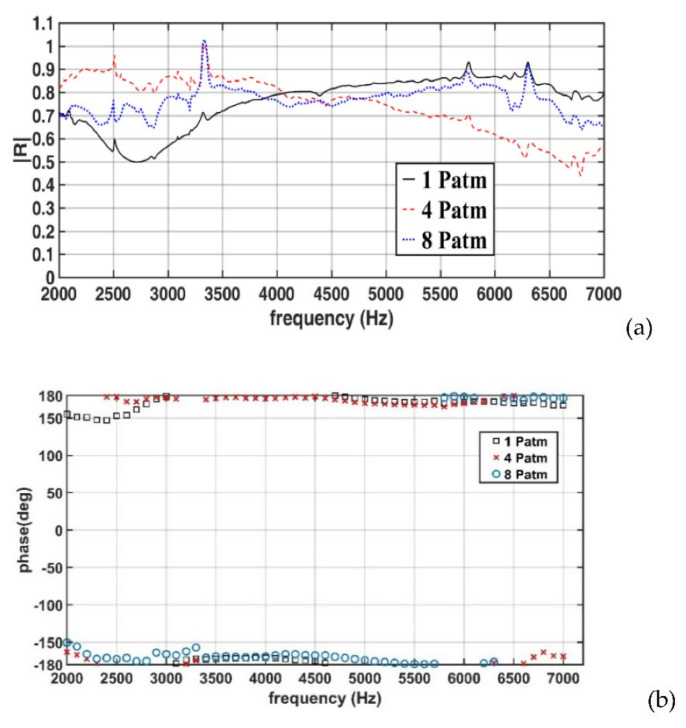
(**a**) Amplitudes and (**b**) phases of the measured reflection coefficients for the 25-mm-thick specimen (Case 1).

**Figure 18 sensors-22-03827-f018:**
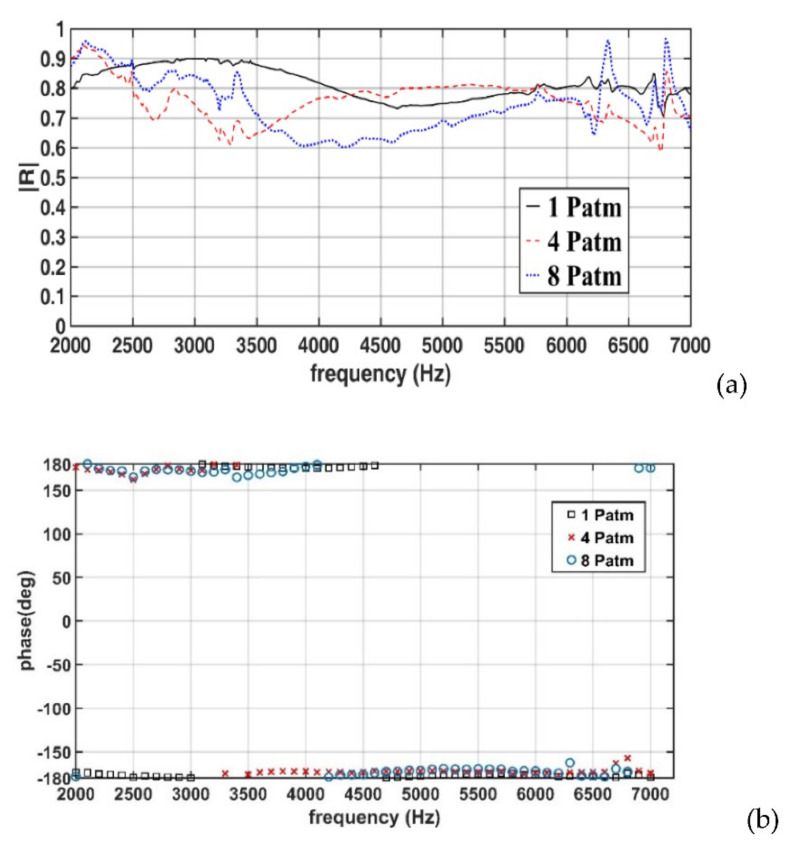
(**a**) Amplitudes and (**b**) phases of the measured reflection coefficient for the 50-mm-thick specimen (Case 2).

**Figure 19 sensors-22-03827-f019:**
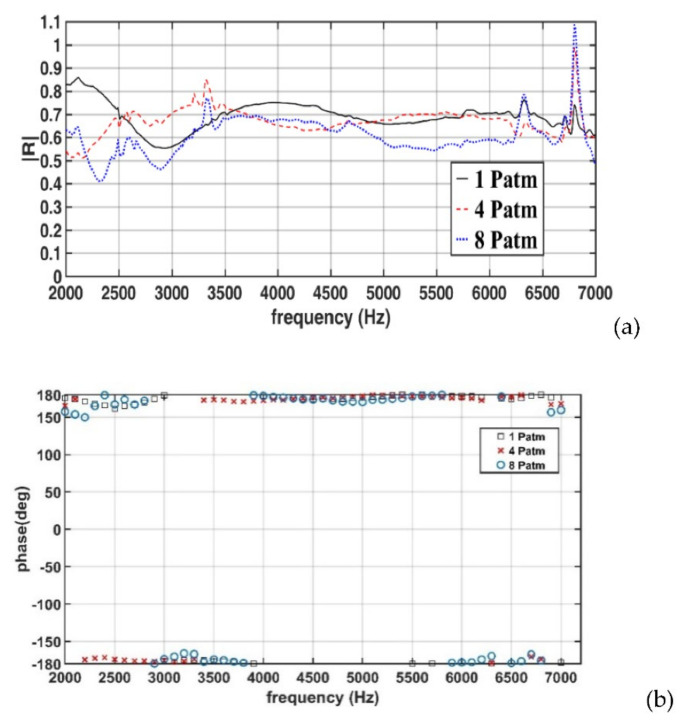
(**a**) Amplitudes and (**b**) phases of the measured reflection coefficients for the 75-mm-thick specimen (Case 3).

**Table 1 sensors-22-03827-t001:** Material and wave parameters of the developed pressurized WFIT.

	Wave Velocity (m/s)	Density (kg/m^3^)	Size (cm)	
Water (20 °C)	$c_{1}=1480$ $c_{eff}=1458$	$\rho_{1}=998.2$	Inner diam.	b=11.6
Tube Wall	$c_{L}=5640$ *	$\rho_{w}=7900$	Outer diam.	d=17.6
	$c_{T}=3070$ *		Length	L=150

* Longitudinal and transverse elastic wave speeds in 304L stainless steel were obtained from ASM International [27].

**Table 2 sensors-22-03827-t002:** LabVIEW parameters of the sound-emitting system.

Sampling frequency	50 kHz
Amplitude of incident sound	10.0 mv
Sound-transmitting period	0.21 s
Signal duration time	0.15 s
Sound frequency	2–7 kHz

**Table 3 sensors-22-03827-t003:** Maximum errors between theoretical reflection coefficients and measured reflection coefficients at the water–air interface when various calibration methods were used.

	Corbett [7]	Wilson et al. [8]	Jian [14]	Present
Calibration method	Sensor-switching	3PCM	Sensor-switching	3PCM
Operating frequency	2–21 kHz	5–9 kHz	1–8 kHz	2–7 kHz
Amplitude error	0.07	0.015	0.11	0.02
Phase error	30°	0.7°	30°	4°

**Table 4 sensors-22-03827-t004:** Values of $\lambda_{i}(i=1,\;2,\;3)$ (calibration parameters) obtained in three tests conducted on different dates.

Test	$\lambda_{1}$	$\lambda_{2}$	$\lambda_{3}$
Fist	1.1991 + 0.6836i	0.9283 − 0.8738i	0.5510 + 1.0913i
Second	1.2208 + 0.7549i	0.9191 − 0.7505i	0.5777 + 0.9940i
Third	1.1044 + 0.7913i	0.9300 − 0.5589i	0.5960 + 0.8583i

**Table 5 sensors-22-03827-t005:** Conditions in the experiments conducted to determine the acoustic properties of the rubber specimens.

Case	Thickness (mm)	Pressure (P_atm_)
Case 1	25	1, 4, 8
Case 2	50	1, 4, 8
Case 3	75	1, 4, 8

## Data Availability

Not applicable.

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
