# Peer review of "Development and Applications of a Pressurized Water-Filled Impedance Tube"

_sensors, 2022, doi:10.3390/s22103827_

Round 1
Reviewer 1 Report
The manuscript presented a very detailed reflection measurement performed by impedance tube. The proposed method was verified by a well-known clean system water-air interface and compared with literature methods showing good performance. The method was then applied to measure the pores rubber samples with different thickness. The study provides valuable technique to the field. However, some points were not clear in the current presentation. The reviewer suggests a revision of the manuscript. The detailed comments as following:
- What absolute(R) refers to? This is confusing. Was the factor reflection coefficient or reflectivity?
- In the rubber measurement, were the reflection coefficients before absolute complex number? It seems like the rubber thickness values were shorter than the operating wavelength. In this case, the sample reflected the wave with under-loading condition. Hence, both reflection coefficient and transmission coefficient should be complex number. The reviewer suggests the authors to include reflection coefficients without absolute.
- With the sample thickness increase approaching to the operating wavelength, the imaginary part of the reflection coefficients. If the un-absolute reflection coefficients were complex, the reviewer suggests including the real part of reflection coefficients, which was expected to increase along the increase of the sample thickness.
- The phase shift in rubber sample was confusing. 180 degrees shift on reflection means a complete out-of-phase wave cancellation which should provide very low measured amplitude of signal at the hydrophone locations. The reviewer suggests some additional discussion on the 180 degrees shift.
- The pressurized conditions should induce a slightly higher speed of sound and effective. The effective impedance and reflection coefficients should increase. The reviewer suggests some additional discussion on that.
- Pulse-echo setups was also a comment category in reflection or impedance measurements. The reviewer suggests an additional paragraph in introduction section including pulse-echo reflection or impedance measurement setups such as:
DOI: 10.1063/1.3296225
DOI: 10.1109/TUFFC.2019.2950343
DOI: 10.1007/s12647-009-0026-6
DOI: 10.1007/s12647-013-0051-3
Author Response
We appreciate the interest that the reviewer has taken in our manuscript and the constructive suggestions he has provided. Our revisions reflect all the reviewer’s suggestions and comments. Detailed responses are provided below.

Reviewer 2 Report
In this manuscript, the author constructs a pressure underwater acoustic tube and the corresponding test system, which can test the reflection coefficient of underwater acoustic material surface under different pressures. I recommend publishing this article, but the author needs to response the following comments:
- Is the pressurization system liquid or gas? In addition, how long does it take to maintain the pressure after pressurization?
- Is the pressurization system liquid or gas? In addition, how long does it take to maintain the pressure after pressurization?
- In the underwater acoustic tube, the water cannot have bubbles, otherwise the bubbles can cause serious deviation to the test results. Since the pipe wall is made of metal, especially in winter, micro bubbles may be generated in the pipe wall. How to avoid this situation?
Author Response

(The authors gave the same response as above.)

Round 2
Reviewer 1 Report
The authors addressed the comments from the reviewer well. The reviewer suggests the acceptance.